# Effects of social determinants on children's health in informal settlements in Bangladesh and Kenya through an intersectionality lens: a study protocol

Eliud Kibuchi [1], Proloy Barua,[2] Ivy Chumo,[3] Noemia Teixeira de Siqueira Filha,[4] Penelope Phillips-Howard,[5] Md Imran Hossain Mithu [2] Caroline Kabaria,[3] Zahidul Quayyum [2] Lana Whittaker,[6] Laura Dean,[6] Ross Forsyth,[1] Tasmiah Selim,[2] Bachera Aktar,[2] Varun Sai,[7] Sureka Garimella,[7] Samuel Saidu,[8] Ibrahim Gandi,[9] Lakshmi K Josyula,[10] Blessing Mberu,[3] Helen Elsey [4] Alastair H Leyland,[1] Linsay Gray[1]

For numbered affiliations see end of article.

**Correspondence to**
Dr Eliud Kibuchi;
Eliud.Kibuchi@glasgow.ac.uk

## ABSTRACT

**Introduction** Several studies have shown that residents of urban informal settlements/slums are usually excluded and marginalised from formal social systems and structures of power leading to disproportionally worse health outcomes compared to other urban dwellers. To promote health equity for slum dwellers, requires an understanding of how their lived realities shape inequities especially for young children 0–4 years old (ie, under-fives) who tend to have a higher mortality compared with non-slum children. In these proposed studies, we aim to examine how key Social Determinants of Health (SDoH) factors at child and household levels combine to affect under-five health conditions, who live in slums in Bangladesh and Kenya through an intersectionality lens.
**Methods and analysis** The protocol describes how we will analyse data from the Nairobi Cross-sectional Slum Survey (NCSS 2012) for Kenya and the Urban Health Survey (UHS 2013) for Bangladesh to explore how SDoH influence under-five health outcomes in slums within an intersectionality framework. The NCSS 2012 and UHS 2013 samples will consist of 2199 and 3173 under-fives, respectively. We will apply Multilevel Analysis of Individual Heterogeneity and Discriminatory Accuracy approach. Some of SDoH characteristics to be considered will include those of children, head of household, mothers and social structure characteristics of household. The primary outcomes will be whether a child had diarrhoea, cough, fever and acute respiratory infection (ARI) 2 weeks preceding surveys.
**Ethics and dissemination** The results will be disseminated in international peer-reviewed journals and presented in events organised by the Accountability and Responsiveness in Informal Settlements for Equity consortium and international conferences. Ethical approval was not required for these studies. Access to the NCSS 2012 has been given by Africa Population and Health Center and UHS 2013 is freely available.

## STRENGTHS AND LIMITATIONS OF THIS STUDY

⇒ These proposed studies will be unique because we will quantitatively provide an understanding of the Social Determinants of Health (SDoH) that drive health inequalities for children under-five years old (0–4 years) living in Nairobi and Dhaka slums within intersectionality framework using Multilevel Analysis of Individual Heterogeneity and Discriminatory Accuracy approach.

⇒ We will use Nairobi Cross-sectional Survey 2012 (NCSS 2012) and Urban Health Survey (UHS 2013) which are one of the few slum surveys in the global south which contain SDoH that shape health inequalities among urban dwellers.

⇒ The passage of time since conduct of NCSS 2012 and UHS 2013: the data are over 9 years old and will need to be interpreted cautiously due to the dynamic nature of slums.

## INTRODUCTION

People in urban informal settlements also known as slums face disproportionally worse health outcomes compared with other urban dwellers.[1] Slum areas are characterised by inadequate access to safe water, inadequate access to sanitation and other infrastructure, poor structural quality of housing, overcrowding and insecure residential status.[2] Health outcomes are shaped by Social Determinants of Health (SDoH)—characteristics in which living takes place.[3] Critically, SDoH help shape social hierarchies that in turn determine the distribution of power, prestige and resources among groups in society.[4] Slum dwellers are usually excluded and marginalised from formal social systems and

structures of power due to legal informality of their dwellings, which denies them rights to access resources, and in turn leads to health inequities.[1 5] Health inequities are differences in health outcomes and in the distribution of health resources experienced between different population groups due to their differences in SDoH.[6] Empowerment of slum dwellers coupled with investments in health systems and infrastructure is required to reduce health inequities particularly among socially vulnerable groups.[7] In turn, this makes empowered slum dwellers accountable because they have increased collective control over the factors that shape their health.

Lack of data that represent the population in slums across cities has been identified as a major hindrance to answering questions critical to the health needs of the slum. In turn, this has created obstacles to understanding the health inequities in slum areas for the effective urban health programming by local governments and other stakeholders.[8 9] Currently, 22.8% of the world's population live in slums, and over 90% of slum dwellers live in low/middle-income countries (LMICs), including hundreds of millions of children.[10 11] Aggregated statistics show that child mortality and health outcomes in rural and urban areas in LMICs have improved between 2000 and 2014.[12] However, during the same period, studies have shown that children in slums tend to experience worse health outcomes than other urban and rural areas.[8 12–16] This is because slums are known to have poor services including water drainage, lack of piped water, flooding, poor sewerage and housing challenges such as overcrowding which are risk factors for waterborne and vector-borne diseases.[17–19] For example, infants who live in slums without piped water may have been shown to experience up to 4.8-fold higher rates of death from diarrhoea.[18] Since the health of children, particularly those under 5 years old (ie, 0–4 years), depends greatly on health and well-being of their mothers and families and the broad SDoH factors of where they live; it is crucial to understand how these factors intersect to create complex and unique positions of vulnerability for children in slums.[17] To fill the knowledge gap and inform such action, we will systematically examine how various SDoH factors intersect with individual factors to affect the health of under-five children living in slums (LMICs), through an intersectionality lens. The health outcomes which will be considered include diarrhoea, cough, fever and acute respiratory infection (ARI) which mostly affect under-five children in slums.[17] For example, poor sanitation and lack of safe drinking water makes diarrhoea a leading cause of death among children aged below 5 years, while fevers, coughs and ARIs are mainly caused by the poor state of housing and overcrowding.[16 17]

SDoH are defined based on the commission for social determinants of health framework which classifies them into structural and intermediary determinants.[3 20] Structural determinants refer to those factors that generate or reinforce stratification in society by exerting a powerful influence on power, prestige and access to resources and thereby influencing people's health.[3] Structural factors include income, age, education, occupation, gender, race/ethnicity, sexuality, disability and social class. On the other hand, intermediary SDoH are those factors which structural determinants operates through to shape health outcomes of individuals and are grouped into four main categories. They are: (1) material circumstances (eg, housing and neighbourhood quality, consumption potential and physical work environment), (2) psychosocial circumstances (ie, relationships, social support and coping styles), (3) behavioural/biological circumstances (ie, nutrition, physical activity, tobacco consumption, alcohol consumption, substance abuse) and (4) health system, particularly access to healthcare. Any attempts to address health inequalities especially among the vulnerable and marginalised must focus on understanding how these multiple SDoH interact with individual factors to shape health inequalities.

An intersectionality lens provides a systematic approach to examining every person's health outcome as fundamentally different from those of others, based on their unique positioning within a web of interacting social determinants.[3 21–24] It assumes that various SDoH interact and change through time to present unique circumstances for individuals or population groups. Therefore intersectionality allows us to account for the complexity of the real world in understanding how different SDoH influence health inequities through marginalisation and privilege in multiplicative and interactional ways based on the lived realities of different groups, without the need to make prior assumptions regarding the importance of one or multiple social categories.[23 25] Quantitatively this can be explored using Multilevel Analysis of Individual Heterogeneity and Discriminatory Accuracy (MAIHDA) approach.[26–29] This approach will enable us to understand the drivers of health inequities in the context of how individual identities interact with SDoH to promote/rectify health inequalities in dynamic ways among under-five living in slums.[22 23]

### Aim
We explore how SDoH influence under-five children health outcomes among dwellers in slums within an intersectionality framework. The findings will inform how individual and social inequities are shaped, and what action can be taken to offset burdens in terms of effective policy and programme development for vulnerable under-five children.

### Objective
The primary analytic objective is to systematically examine how various SDoH, and individual factors affect health outcomes (ie, diarrhoea, fever, cough and ARI) of under-five children in slums within an intersectionality framework.

### DATA
In the proposed study, separate analyses and papers applying the same statistical methods for Bangladesh

and Kenya are planned. The underlying social and living conditions in Dhaka and Nairobi slums are different which necessitates two distinct publications for Bangladesh and Kenya for effective interventions.[8 30–32] These studies will use cross-sectional data from the Nairobi Cross-section Slums Survey (NCSS 2012) for Kenya and Urban Health Survey 2013 (UHS 2013) for Bangladesh, as these are the current disaggregated datasets for slum surveys in both Bangladesh and Kenya. Disaggregated datasets for secondary data analyses are not available in other countries (India and Sierra Leone), where field activities for Accountability and Responsiveness in Informal Settlements for Equity (ARISE) project are also being implemented.

The choice of the SDoH characteristics to be included as explanatory variables in the planned analyses will be informed by the literature on the factors that influence health outcomes for under-five.[12 17 33–38] These variables include age and sex for under-fives which have been shown to be determinants of childhood morbidity.[12 34] Moreover, under-five health outcomes are closely related to the structural factors such as age and education of head of households and mothers since they affect their ability to provide safe places to grow and live and the ability of households to adopt preventive strategies at a given time.[17 35 37] The poor hygiene practices in slums which are associated with poor water drainage, inadequate access to safe water, open sewers and overcrowding also exacerbates health outcomes for under-five.[39] Finally, higher levels of malnutrition and lower immunisation coverage among under-five living in slums leads to their poor health.[33 40–43]

### Nairobi Cross-section Slums Survey (NCSS 2012)

The NCSS 2012 data were collected by the African Population and Health Research Center from all slums in Nairobi between June and November 2012.[8] The sample to be included in the survey was calculated based on the percentage of children 12–23 months who had been fully immunised using a margin of error of 0.03, design effect of 1.50 and critical value of $\alpha=0.05$.[8] The number of households required to estimate the percentage of children 12–23 months fully vaccinated was large enough to allow estimation of the other indicators such as diarrhoea, fever and cough within the specified precision. A two-stage random sampling methodology was used and a total of 5490 households and 4420 women aged 12–49 years were successfully interviewed yielding a response rate of 88% and 86%, respectively. We will be interested in women because their questionnaire contained a module on child's health, where we are to obtain our health outcomes of interest. Participation was voluntary and no compensation or financial incentive was offered. The 4420 women participants provided data on 2199 children aged five and under.

In this study we will consider three health outcomes for children: (1) whether a child had fever or not, (2) whether a child had cough or not and (3) whether a

child had diarrhoea or not. The predictor variables to be considered in the analysis will be classified into four categories: (1) children's demographics (ie, age and sex), (2) head of household characteristics (ie, sex, ethnic group, education and age), (3) child's mother characteristics (ie, age) and (4) social structure characteristics in the household (ie, wealth index, length of stay, religion, education, tenure, food availability, health insurance, income generating activity, disability and catastrophic health costs). Catastrophic health expenditure will be computed using the empirical methodological procedure used by Buigut et al[44] and we will take a 40% threshold which is informed by Xu et al.[45] The wealth index was generated using source of drinking water, type of toilet facility, cooking fuel used, lighting type at night, material used to construct floor, wall and roof of dwelling, and household possessions (ownership of household items).[8] Detailed description of health outcomes and predictor variables for NCSS 2012 are found in online supplemental table s1.

### Bangladesh Urban Health Survey (UHS 2013)

The UHS 2013 is a representative cross-sectional household survey implemented jointly by (1) National Institute of Population Research and Training (NIPORT), (2) Measure Evaluation, University of North Carolina at Chapel Hill, USA, (3) International Centre for Diarrhea Disease Research, Bangladesh and (4) Associates for Community and Population Research.[32] The survey collected information designed to examine intra-urban differentials in health and service utilisation from 53 790 households. These households were selected using a stratified three-stage sampling procedure in three urban domains: (1) city corporation slum, (2) city corporation non-slum and (3) other urban areas. The key indicators used to calculate the sample size were (1) under-five mortality and (2) percentage of birth deliveries in the health facilities for all births in the last 3 years.[32] Participation was voluntary and no compensation or financial incentive was offered. The proposed analysis will only include the domain of city corporation slum since our interest involves investigating social processes which drive health inequalities in slums. In addition, we will consider the women subsample from the survey because their questionnaire contained a module on child's health and nutrition. The number of households selected in the domain of city corporation slum were 15 750 and those interviewed 14 806 yielding a response rate of 94%. A total of 14 702 women were eligible for interview and 14 011 were interviewed yielding a 95% response rate.

We will consider three child health conditions: (1) whether a child had fever or not, (2) whether a child had cough or not and (3) whether a child had an ARI or not. ARI is a cough accompanied by short, rapid or difficult breathing which is chest related and usually considered as a proxy for pneumonia. The predictor variables to be considered in the analyses will be classified into four categories: (1) children's demographics (ie, age and sex), (2) mother's demographics (ie, religion, age, highest level

**Table 1**  List of outcome and explanatory variables for both Nairobi Cross-sectional Survey (NCSS) 2012 and Bangladesh Urban Health Survey (UHS) 2013

|  | NCSS 2012 Variables | UHS 2013 Variables |
|---|---|---|
| **Outcomes** | | |
| Health outcomes | Diarrhoea, fever, cough | Fever, cough, acute respiratory infection (ARI) |
| **Predictors** | | |
| Under-five demographics | Age, sex | Age, sex |
| Head of household characteristics | Age, sex, education, ethnic group | Age, sex, education, marital status |
| Child's mother characteristics | Age | Age, marital status, ever attended school, highest education, employment |
| Social structural characteristics | Wealth index, length of stay, religion, income generating activity, tenure, disability, food availability, health insurance and health catastrophic costs | Wealth index, dwelling ownership, land ownership, garbage disposal, cooking fuel, having kitchen, migration status, housing type and division |

of education, employment status, ever attended school and marital status), (3) head of household demographics (ie, sex and age) and (4) social structure characteristics in the household (ie, wealth index, dwelling ownership, land ownership, cooking fuel, garbage disposal method, kitchen type, house type and division). Wealth index was constructed by data provider using principal components analysis based on the following variables: dwelling characteristics such as presence of electricity, type of water source, type of toilet, and floor, wall, and roof material, household ownership of selected assets and durable goods (radio, television, motorcycle, computer, refrigerator, electric fan and automobile) and two indicators of housing tenure (whether the household held title to the dwelling and/or the land). A detailed description of variables and their categorical levels for UHS 2013 are presented in online supplemental table s2.

Table 1 presents a summary of variables which will be considered for analyses for NCSS 2012 and UHS 2013. The differences in outcome and predictor variables in NCSS 2012 and UHS 2013 also informed the need for separate analyses for Kenya and Bangladesh. Data on ARI and diarrhoea were not available in the NCSS 2012 and UHS 2013, respectively. A causal diagram showing the direct pathway between the four categories of variables and under-five health outcomes is shown in figure 1.

## STATISTICAL METHODS

The effects of SDoH on children's health outcomes in slums through an intersectionality lens will be assessed using MAIHDA approach developed to analyse intersectional inequalities.[26–28 46] MAIHDA aims to primarily identify intersecting inequalities in a quantitative way by defining intersectional groups according to combinations of social attributes which is like clustering of individuals based on some shared attributes such as neighbourhood, school or household, among others.[27 47] That is, individuals can be clustered based on abstract groupings such as

a set of SDoH associated with their intersectional social identities and individual characteristics.

MAIHDA therefore, allows multiplicative modelling of health inequalities at the intersection of multiple SDoH by analysing the heterogeneity (ie, differences) within and between intersectional groups/strata by separating variance (ie, the measure of variation) into—the between-strata (ie, differences across strata) and the within—strata (ie, differences of individuals within a given stratum).[27 28]

The advantage of MAIHDA is that we will look at intersectionality as a mix of both marginalisation and privilege.[26 27] Generally, an interaction-based fixed effects approach looks at intersectionality from the perspective of marginalisation only, which runs the risk of reinforcing the notion of social dominance of the privileged groups which are used as 'default' categories. In addition, from an analytical perspective, MAIHDA models do not face the issues of scalability (ie, a model's inability to accommodate an increase in the number of variables included), model parsimony (ie, a simple model not having great explanatory predictive power) and reduced sample size in some intersectional groups (which influences whether an effect size is determined or not).[27] If desired, we can

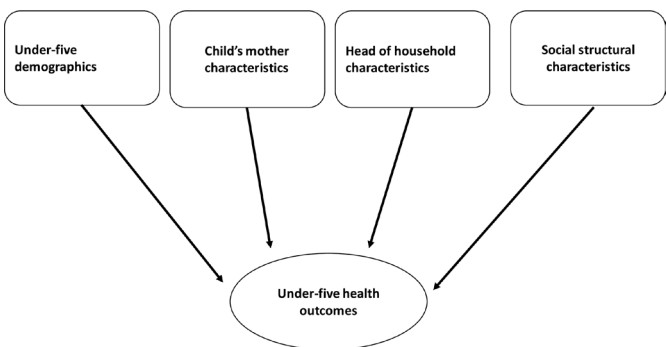

**Figure 1**  A causal diagram showing the direct pathways between children's demographics, child's mother characteristics, head of household demographics, social structure characteristics and under-five health outcomes.

extend this multilevel framework applied in MAIHDA into a multivariate multilevel model to analyse more than two health outcomes simultaneously, especially for health outcomes which occur concurrently affecting other aspects of life.[48 49] MAIHDA estimates intersectional effects in two steps.[27 47]

First, a null model (ie, model 1) will be specified with individuals at level one nested within social groups at level two to assess whether there is significant clustering within intersectional strata. Model 1 will not include any predictor variables and will only have an intercept to estimate the mean health outcome and a random effect to model intersectional strata differences (ie, variance). This will allow estimation of the extent to which the variance in an outcome is explained by differences across intersections vs differences within using variance partitioning coefficient (VPC) also known as intraclass coefficient (ICC) and the area under the receiver operating characteristic curve (AUC-ROC).[26 27 47] VPC and AUC-ROC measures will also be used to quantify discriminatory accuracy (DA) which is ability of the model to discriminate individuals with and without an outcome of interest.[26 27 29 50] VPC quantifies the share of the total individual variance in having an outcome that is accounted for at the intersectional strata level with values higher than 5% indicating an acceptable DA.[27 46 51] That is, a high VPC indicates that intersections have a substantially different mean levels of an outcome and that individuals within this group are similar, while a low VPC indicates that individuals within an intersectional group differ substantially.[46 50] On the other hand, AUC-ROC measures the ability of the model to classify individuals with or without an outcome as a function of individual's predicted probabilities and is and is bounded between 0.5 and 1[51 52] (see online supplemental file for details).

In the second step, we will extend model 1 by adjusting for variables used in constructing intersectional strata as fixed effects (ie, model 2). Model 2 will be used to explore to which extent intersectional strata differences will be explained by SDoH used in constructing intersectional groups. Fixed effects in model 2 will be used to estimate model regression coefficients which will be presented as odds ratio (OR) and will describe the association between SDoH variables and under-five health outcomes. In the absence of intersectional strata differences, fixed effects used to construct strata are expected to completely explain intersectional strata differences obtained in model 1. That is, VPC values will be around zero in model 2 indicating that intersectional effects are fully explained by fixed effects and are therefore additive and not multiplicative.[28 46] This will indicate absence of any stratum/group specific interactions since the fixed effects used to construct intersectional strata will completely explain the between stratum variance and all stratum random effects.

However, if strata random effects in model 2 are not equal to zero and assuming no relevant variables are omitted on the model it will indicate existence of multiplicative intersectional effects. This will imply that certain intersectional groups are more vulnerable to an outcome of interest compared with other groups. To assess the proportion of variance explained by the adding fixed effects in model 2 we will compute the proportional change in variance (PCV) of intersectional strata between models 1 and 2 (see more details online supplemental file). The lower the PCV, the higher the amount of unexplained variance which can be due to either interaction effects or omitted variables in the model. Detailed description of model 1, model 2, VPC and PCV are provided in online supplemental file.

These models 1 and 2 can even be extended to include more than two health outcomes in a multivariate multilevel model.[48 49] This model explicitly evaluates the covariance (ie, a measure of joint variability of two random variables) between different social strata and health outcomes which allows us not only to draw conclusions about social group-specific differences but also correlations between health outcomes.[48] This is desirable since we will assume that they capture related, though distinct, health constructs.

Limitation of this proposed studies is that datasets which will be used were collected over 9 years ago.[8 32] Considering the dynamic nature of slums, a more recent data would have been more informative of the SDoH factors which affect under-five health conditions. Despite this, we expect the findings which will obtained to be of great value since these datasets come from the most recently conducted slum surveys in both Bangladesh and Kenya.

## Patient and public involvement

There will be no patient or public involvement in this study, as it is based on secondary data.

## State date of the analyses

September 2021.

## Anticipated end date

March 2022.

## ETHICS AND DISSEMINATION

The study will use secondary data from the Nairobi Cross-sectional Survey 2012 (NCSS 2012) and Urban Health Survey (UHS 2013) which excludes any participant identifiers. Ethical approval for the NCSS 2012 study was obtained from the Kenya Medical Research Institute's Ethics Review Committee.[8] For UHS 2013, ethical approval was obtained from the Bangladesh Medical Research Council and the Institutional Review Board at the School of Public Health, University of North Carolina at Chapel Hill.[32] This work as part of ARISE will be used in shaping actions to improve slum health for under five in Bangladesh and Kenya.[53] Finding from these studies will be in published peer-reviewed journals and presented in international conferences. Analyses will be presented to policy makers and stakeholders of slum health throughout the course of ARISE project.

**Author affiliations**
[1]MRC/CSO Social and Public Health Sciences Unit, University of Glasgow, Glasgow, UK
[2]School of Public Health, BRAC University James P Grant School of Public Health, Dhaka, Bangladesh
[3]African Population and Health Research Center, Nairobi, Kenya
[4]Department of Health Sciences, University of York, York, UK
[5]Department of Clinical Sciences, Liverpool School of Tropical Medicine, Liverpool, UK
[6]Department of International Public Health, Liverpool School of Tropical Medicine, Liverpool, UK
[7]The George Institute for Global Health India, New Delhi, Delhi, India
[8]COMAHS, Freetown, Western Area, Sierra Leone
[9]Centre Of Dialogue On Human Settlement And Poverty Alleviation (CODOHSAPA), Freetown, Sierra Leone
[10]The George Institute for Global Health, Hyderabad, India

**Contributors** EK, LG and AHL conceived and designed the study. EK wrote the first draft and revised the protocol. PB, IC, NTdSF, PP-H, MIHM, CK, ZQ, LW, LD, RF, TS, BA, VS, SG, SS, IG, LKJ, BM, HE, AHL and LG critically revised the manuscript.

**Funding** The GCRF Accountability for Informal Urban Equity Hub ('ARISE') is a UKRI Collective Fund award with award reference ES/S00811X/1. The MRC/CSO Social and Public Health Sciences Unit is funded by the Medical Research Council (MC_UU_00022/2) and the Scottish Government Chief Scientist Office (SPHSU17).

**Disclaimer** The funders have no role in study design or writing of the report. The views expressed in this article are those of the authors.

**Competing interests** None declared.

**Patient and public involvement** Patients and/or the public were not involved in the design, or conduct, or reporting, or dissemination plans of this research.

**Patient consent for publication** Not applicable.

**Provenance and peer review** Not commissioned; externally peer reviewed.

**ORCID iDs**
Eliud Kibuchi http://orcid.org/0000-0002-5091-6450
Md Imran Hossain Mithu http://orcid.org/0000-0001-8057-3092
Zahidul Quayyum http://orcid.org/0000-0002-2276-4576
Helen Elsey http://orcid.org/0000-0003-4724-0581

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
