## [Reviewer comments · BMJ Open]

ARTICLE DETAILS

TITLE (PROVISIONAL)	Effects of social determinants on children's health in informal settlements in Bangladesh and Kenya through an intersectionality lens – a study protocol
AUTHORS	Kibuchi, Eliud; Barua, Proloy; Chumo, Ivy; Teixeira de Siqueira Filha, Noemia; Phillips-Howard, Penelope; Mithu, Md. Imran Hossain; Kabaria, Caroline; Quayyum, Zahidul; Whittaker, Lana; Dean, Laura; Forsyth, Ross; Selim, Tasmiah; Aktar, Bachera; Sai, Varun; Garimella, Sureka; Saidu, Samuel; Gandi, Ibrahim; Josyula, Lakshmi; Mberu, Blessing; Elsey, Helen; Leyland, Alastair; Gray, Linsay

VERSION 1 – REVIEW

REVIEWER	Macharia, William The Aga Khan University, Paediatrics
REVIEW RETURNED	21-Sep-2021

GENERAL COMMENTS	A well thought out and written paper save for where under DATA section in Methods tenses are confusing - past in reference to the source data and present in reference to the proposed study.
---

REVIEWER	Zinszer, Kate Universite de Montreal Institut de recherche en sante publique, Department of Social and Preventive Medicine
REVIEW RETURNED	15-Oct-2021

GENERAL COMMENTS	I was quite interested in reading more about the study after the excellent introduction but then was disappointed in terms of the scope of the study, which is a secondary analysis based on data from 2012 and 2013. A study protocol is not necessary for the scope of the proposed work, it could be easily summarized in a publication. A protocol is appropriate when there is a need for timely dissemination of study designs or for the prevention of study duplication (which the latter is a non-issue given the dates associated with the datasets).
---

REVIEWER	Reid, Graham University of Western Ontario, Psychology & Family Medicine
REVIEW RETURNED	10-Feb-2022

GENERAL COMMENTS	Title: Effects of social determinants on children's health in informal settlements in Bangladesh and Kenya through an intersectionality lens – a study protocol BMJopen-2021-056494
--

Comments: This paper describes the data analytic plan for examining intersectionality in relation to health outcomes among young children (0-4 years) living in urban informal settlements (i.e., slums) in Nairobi and Bangladesh. Secondary data analyses are proposed for the Nairobi Cross-sectional Slum Survey (NCSS) 2012 for Kenya and the Urban Health Survey (UHS) 2013 for Bangladesh. The aim is to understand how the social determinants of health intersect with individual factors that shape intersecting inequities to create unique positions of vulnerability for children. Multilevel Analysis of Individual Heterogeneity and Discriminatory Accuracy (MAIHDA) will be used to model this complexity.

The proposed analyses seek to examine a challenging and important research question. Issues of equity are clearly important to address, and the impact of inequities should be clearly evident in the marginalized population of children living in slums.

However, there are concerns with this manuscript that should be addressed. Concerns are summarized below under overall issues and under specific subsections of the manuscript, identified as either "Major" or "Minor" concerns.

**ABSTRACT
MINOR**

"urban informal settlements" - might add "(i.e., slums)" after to link to subsequent use of the term

"especially among under-fives ..." – I think this would read clearer to just say "especially for young children (0-4 years old)."

Change - multilevel analysis of individual heterogeneity and discriminatory accuracy (MAIHDA) to Multilevel Analysis of Individual Heterogeneity and Discriminatory Accuracy (MAIHDA)

Consider describing the sample and what is being assessed. It is hard to understand data analyses without know what the data are.

"to understand how social determinants"- consider adding examples of the social determinants being assessed

Might be clearer to change "NCSS (2012)" to NCSS 2012; similarly, for UHS (2013)

INTRODUCTION Major

"Aspects of socio-economic status include income, age, education, occupation, gender, sex, race/ethnicity, sexuality, disability, and social class." – This is an unusual way of framing socio-economic status. The authors should examine the literature to find a citation to support this conceptualization. Typically, socioeconomic status refers to educational attainment, occupation, and/or family income.

In the paragraph "An intersectionality lens provides...." the sentence "This is achieved by using multilevel analysis of individual heterogeneity...." seems out of place. I think this should be removed, as this is better placed in the data analyses section.

A recent citation that might be helpful is Homan, P., et al. (2021). "Structural intersectionality as a new direction for health disparities research." J Health Soc Behav 62(3): 350-370.

	The last paragraph on the ARISE consortium seems disconnected with the rest of the introduction. Perhaps this could be placed at the end of the manuscript in a new subsection on implications of the project. What appears to be missing from the introduction is a review the literature supporting what is being measured in the survey. What is the rationale for the choice of variables being measured in relation to the literature on health equity? In the Objectives subsection, what “health conditions” refers to should be made explicit, with the rationale for the choices of health conditions to measure reviewed in the introduction. Some of this material appears to be in the Data subsection but might be better placed in the introduction. Minor Define what an “urban informal settlements” refers to; maybe just add this as the second sentence in the intro. It would be helpful to define or give examples of what you mean by “formal social systems and structures of power”. Define what you mean by “health equity”. “slum dwellers (who constitute most of city dwellers)” – this is unclear. I don’t think the authors mean that most of the people who live in a city live in slums. “Despite aggregated statistics showing improved mortality and health outcomes in urban...” – clarify the context for “improved”; if this refers to changes over time, cite the years. In the paragraph on social determinants, it would be helpful to give examples of “political institutions” (line 46). METHODS & ANALYSES Major Describing secondary data analyses is a challenge and there is not standard approach in the literature. The authors should consider describing the methods for each of the primary surveys, followed by variables specific to their analyses/study. Separate analyses and papers for the two Bangladesh and Nairobi surveys are planned. A rationale for two separate papers should be provided. It may be helpful to provide a table showing all the variables from both surveys, which may highlight differences supporting separate analyses. Each of the variables should be described in detail. For example, how is the wealth quintile index actually computed? What is support for this approach to computing a composite score? When describing the Bangladesh Urban Health Survey, they state that a principal components analysis (PCA) will be conducted. Again, the rationale and previous use of this approach should be justified. PCA tends to yield samplespecific weights that may or may not replicate across samples. The exact method of computing the composite score should be described and whether PCA weights or unit weighting will be used.
--	---

	Sample inclusion and exclusion criteria for the proposed project/analyses should be clearly articulated. A related point is whether or not any compensation was provided to participants. Including conceptual models is a strength. The authors have included a figure/diagram for their analyses. However, the diagram does not capture the complexities of intersectionality outlined in the introduction. There are no indications of what the interactions across levels are, nor are potential mechanisms shown. Mechanisms are described: eg “their higher prevalence is caused by poor characteristics of physical environment such as poor water drainage, inadequate access to safe water, open sewers, and overcrowding..” However, the figure and analyses are not capturing these mechanisms. The MAIHDA approach is clearly sophisticated modelling needed to capture the complex phenomena inherent in intersectionality. The authors have a full description in the appendix appropriate to the statistician. It is a challenge to make the description accessible for the non-statistician. I think the text could be improved by a more step-by-step approach to model building and how to interpret the findings. For example, why is discriminatory accuracy important and what does it tell us in relation to the substantive research questions? The description of the MAIHDA approach that is hard to follow is the between vs within level interactions. The description appears to be looking at all multi-way interactions for all predictors. What is the theoretical rationale for testing all interactions vs specific within and between level interactions? Finally, it would be important to tie the analyses back to issues of inequity outlined in the introduction. A particular challenge may be understanding specific inequities within a population for whom all individuals are marginalized. Minor “A total of 2,199 children’s data aged under 5 years was provided in the women’s questionnaire.” – should be “data....were”. ETHICS A statement related to ethics approval should be provided. REFERNCES A number of references appear to be missing elements. I saw the following, but all references should be reviewed. 18. Organization WH. A conceptual framework for action on the social determinants of health. 2010. 19. Green MA, Evans CR, Subramanian SV. Can intersectionality theory enrich population health research? 2017. 20. López N, Gadsden VL. Health inequities, social determinants, and intersectionality. 2016. 23. . !!! INVALID CITATION !!! (11, 12).
--	--

	Also see the appendix {Merlo, 2018 #375}
--	--

REVIEWER	Metzler, Janna Columbia University, Heilbrunn Department of Population and Family Health
REVIEW RETURNED	15-Feb-2022

GENERAL COMMENTS	Dear Authors, I want to thank you for contributing this protocol to the literature and further examining health disparities to support the health and wellbeing of children in urban informal settlements. I've highlighted below a few areas for further clarification that I found challenging when reading through the protocol. Objective Indeed, an intersectional lens on disease burden will lend itself to a more holistic understanding of how to intervene within the population. However, to me, the intent of the analysis is to also provide recommendations for how to intervene to reduce incidence of health conditions noted in the data section. The connection to practice is relevant for readers and can be considered by the authors for its inclusion. Data It would be helpful to clarify why NCSS 2012 and UHS 2013 household survey data were not used or relevant to the analysis. Further explanation as to why other predictors were not included in the analysis may also be relevant for readers – including mother's age of first marriage (where there is evidence on child marriage as being interconnected to their offspring's health outcomes); marital status or husband occupancy in the household connected to female-headed household vulnerabilities; etc.
--

VERSION 1 – AUTHOR RESPONSE

Reviewer: 1

Dr. William Macharia, The Aga Khan University Comments to the Author:

- A well thought out and written paper save for where under DATA section in Methods tenses are confusing - past in reference to the source data and present in reference to the proposed study.
- ✓ We are going to use secondary data in the proposed studies and that is the reason why some sentences are in past tense in data section. The Nairobi Cross-section Slums Survey data was collected in 2012 while Bangladesh Urban Health Survey was collected in 2022.

Reviewer: 2

Dr. Kate Zinszer, Universite de Montreal Institut de recherche en sante publique Comments to the Author:

- I was quite interested in reading more about the study after the excellent introduction but then was disappointed in terms of the scope of the study, which is a secondary analysis based on data from 2012 and 2013. A study protocol is not necessary for the scope of the proposed

work, it could be easily summarized in a publication. A protocol is appropriate when there is a need for timely dissemination of study designs or for the prevention of study duplication (which the latter is a non-issue given the dates associated with the datasets).

✓ Protocols are necessary for secondary data analyses when undertaking different analyses using same methodologies in different settings. This ensures that same procedures are followed while ensuring equitable partnership across different partners. For example, in our case of ARISE Hub we have partners from five different countries, UK, Bangladesh, Sierra Leone, Kenya, and India. One of the core objectives of ARISE is to promote equitable partnership and capacity building across partners in different countries. The Kenya analyses will be led by the main author of this protocol who is based in UK. On the other hand, Bangladesh team will lead their analyses with the support from the main author as a way of capacity building. This necessitated the need for a protocol which will be used by both teams while undertaking their separate analyses.

Reviewer 3

Dr. Graham Reid, University of Western Ontario Comments to the Author:

Please see attached file

ABSTRACT

MINOR

- “urban informal settlements” - might add “(i.e., slums)” after to link to subsequent use of the term “especially among under-fives ...” – I think this would read clearer to just say “especially for young children (0-4 years old).”
- ✓ Several studies have shown that residents of urban informal settlements/slums are usually excluded and marginalised from formal social systems and structures of power leading to disproportionately worse health conditions compared to other urban dwellers.
- ✓ To promote health equity for slum dwellers, requires an understanding of how their lived realities shape inequities especially for young children 0-4 years old (i.e., under-fives) who tend to have a higher mortality compared with non-slum children.
- Change - multilevel analysis of individual heterogeneity and discriminatory accuracy (MAIHDA) to Multilevel Analysis of Individual Heterogeneity and Discriminatory Accuracy (MAIHDA)
- ✓ We will apply Multilevel Analysis of Individual Heterogeneity and Discriminatory Accuracy (MAIHDA) to explore multiple interactions in an intersectionality perspective on children’s health conditions
- Consider describing the sample and what is being assessed. It is hard to understand data analyses without know what the data are.
- ✓ The analysis sample will be based on complete case analyses with variables to be included in the analyses will be selected via univariate analyse.
- “to understand how social determinants”- consider adding examples of the social determinants being assessed.
- ✓ Some of SDoH characteristics to be considered will include child’s age and sex, head of household characteristics such as sex and age and social structure characteristics of household such as wealth index, residence status among others
- Might be clearer to change “NCSS (2012)” to NCSS 2012; similarly, for UHS (2013)
- ✓ The protocol describes how we will analyse data from the Nairobi Cross-sectional Slum Survey (NCSS 2012) for Kenya and the Urban Health Survey (UHS 2013) for Bangladesh.

INTRODUCTION

Major

- “Aspects of socio-economic status include income, age, education, occupation, gender, sex, race/ethnicity, sexuality, disability, and social class.” – This is an unusual way of framing socio-economic status. The authors should examine the literature to find a citation to support this conceptualization. Typically, socioeconomic status refers to educational attainment, occupation, and/or family income.
- ✓ This has been updated to read: SDoH are defined based on the commission for social determinants of health (CDSH) framework which classifies them into structural and intermediary determinants (3, 21, 22). Structural determinants refer to those factors that generate or reinforce stratification in society by exerting a powerful influence on power, prestige, and access to resources and thereby influencing people’s health (22). Structural factors include income, age, education, occupation, gender, race/ethnicity, sexuality, disability, and social class. On the other hand, intermediary SDoH are those factors which structural determinants operates through to shape health outcomes of individuals and are grouped into four main categories.
 - In the paragraph “An intersectionality lens provides....” the sentence “This is achieved by using multilevel analysis of individual heterogeneity....” seems out of place. I think this should be removed, as this is better placed in the data analyses section.
- ✓ We are also contributing methodologically by applying MAIHDA approach in slum health studies which necessitated the need to mention the approach in this section.
 - A recent citation that might be helpful is Homan, P., et al. (2021). "Structural intersectionality as a new direction for health disparities research." J Health Soc Behav 62(3): 350-370.
- ✓ We have included this as a reference inequality. However, the statistical approach applied in this paper is interaction -based fixed effects mode which is different from MAIHDA approach. Their random effects accounts for differences attributed at state level. However, in our analyses random intercepts will account for differences in intersectional strata created using social determinant of health.
 - The last paragraph on the ARISE consortium seems disconnected with the rest of the introduction. Perhaps this could be placed at the end of the manuscript in a new subsection on implications of the project.
- ✓ We have removed that paragraph entirely.
 - What appears to be missing from the introduction is a review the literature supporting what is being measured in the survey. What is the rationale for the choice of variables being measured in relation to the literature on health equity?
- ✓ This paragraph explains supports the choice of the variables to be included in the analyses: The choice of the SDoH characteristics to be included as explanatory variables in the planned analyses will be informed by the literature on the factors that influence health conditions for under-five and discussions with researchers from where the data were collected (12, 20, 23-28). These variables include age and sex for under-fives which have been shown to be determinants of childhood morbidity (36). Moreover, under-five health conditions are closely related to the structural factors such as age and education of head of households and mothers since they affect their ability to provide safe places to grow and live and the ability of households to adopt preventive strategies at a given time (17, 36). The poor hygiene practices in slums which are associated with poor water drainage, inadequate access to safe water, open sewers, and overcrowding also exacerbates health conditions for under-five (29). Finally, higher levels of malnutrition and lower immunisation coverage among under-five living in slums leads to their poor health (23, 30-33).

- In the Objectives subsection, what “health conditions” refers to should be made explicit, with the rationale for the choices of health conditions to measure reviewed in the introduction. Some of this material appears to be in the Data subsection but might be better placed in the introduction.
- ✓ Revised to read: “The primary analytic objective is to systematically examine how various SDoH, and individual factors affect health conditions (i.e., diarrhea, fever, cough, and ARI) of under-five children in slums within an intersectionality framework.”

Minor

- Define what an “urban informal settlements” refers to; maybe just add this as the second sentence in the intro.
- ✓ Included: “Slum areas are characterised by inadequate access to safe water, inadequate access to sanitation and other infrastructure, poor structural quality of housing, overcrowding, and insecure residential status”.
- It would be helpful to define or give examples of what you mean by “formal social systems and structures of power”.
- ✓ We have expanded the sentence to explain the statement to read: Slum dwellers are usually excluded and marginalised from formal social systems and structures of power due to legal informality of their dwellings, which denies them rights to access resources, which leads to health inequities
- Define what you mean by “health equity”. “slum dwellers (who constitute most of city dwellers)” – this is unclear. I don’t think the authors mean that most of the people who live in a city live in slums.
- ✓ We have added: “Health inequities are differences in health outcomes and in the distribution of health resources experienced between different population groups due to their differences in SDoH”.
- “Despite aggregated statistics showing improved mortality and health outcomes in urban...” – clarify the context for “improved”; if this refers to changes over time, cite the years.
- ✓ The sentence now reads: Aggregated statistics show that child mortality and health outcomes in rural and urban areas in low and middle income countries have improved between 2000 and 2014.
- In the paragraph on social determinants, it would be helpful to give examples of “political institutions” (line 46).
- ✓ We have rewritten that paragraph into: “SDoH are defined based on the commission for social determinants of health (CDSH) framework which classifies them into structural and intermediary determinants (3, 21, 22). Structural determinants refer to those factors that generate or reinforce stratification in society by exerting a powerful influence on power, prestige, and access to resources and thereby influencing people’s health (22). Structural factors include income, age, education, occupation, gender, race/ethnicity, sexuality, disability, and social class. On the other hand, intermediary SDoH are those factors which structural determinants operates through to shape health outcomes of individuals and are grouped into four main categories. They are: 1) material circumstances (e.g., housing and neighborhood quality, consumption potential, and physical work environment), 2) psychosocial circumstances (i.e., relationships, social support, and coping styles), 3) behavioral/biological circumstances (i.e., nutrition, physical activity, tobacco consumption, alcohol consumption, substance abuse), and 4) health system, particularly access to health care. Any attempts to address health inequalities especially among the vulnerable and marginalised must focus on understanding how these multiple SDoH interact with individual factors to shape health inequalities”

METHODS & ANALYSES

Major

- Describing secondary data analyses is a challenge and there is not standard approach in the literature. The authors should consider describing the methods for each of the primary surveys, followed by variables specific to their analyses/study.
- ✓ We have described the survey design of both the Nairobi Cross-section Slums Survey (NCSS 2012) and Bangladesh Urban Health Survey (UHS 2013) in data section. In addition, we have included the variables to be considered in the analyses as summarised in Figures 1 and 2 and Table 1.

- Separate analyses and papers for the two Bangladesh and Nairobi surveys are planned. A rationale for two separate papers should be provided. It may be helpful to provide a table showing all the variables from both surveys, which may highlight differences supporting separate analyses.
- ✓ The decision for separate analyses is informed by these two reasons as indicated in the document: In the proposed study, separate analyses and papers applying the same statistical methods for Bangladesh and Kenya are planned. This is informed by the differences in health priorities and contexts across cities (i.e., Dakar and Nairobi) resulting in two distinct publications for Bangladesh and Kenya.

The differences in outcome and predictors variables in NCSS 2012 and UHS 2013 also informed the need for separate analyses for Kenya and Bangladesh.

- ✓ We have included Table 1 which presents the outcome and predictor variables for both NCSS 2012 and UHS 2013.

- Each of the variables should be described in detail. For example, how is the wealth quintile index actually computed? What is support for this approach to computing a composite score? When describing the Bangladesh Urban Health Survey, they state that a principal components analysis (PCA) will be conducted.
- ✓ We have explained these composite scores based on the information available from survey reports since the data were provided with wealth index as a variable (i.e., was already computed). The reason to state the variables used in wealth index computation is important since these are factors constitute SDoH which drive health inequalities in slums. Therefore, these variables being in wealth index indicates that we have accounted for them in the model as part of the composite index.
- ✓ The only variable which we will compute is catastrophic health costs using variables provided in NCSS 2012 and we have included references of the approach we are going to apply.
- Again, the rationale and previous use of this approach should be justified. PCA tends to yield sample specific weights that may or may not replicate across samples. The exact method of computing the composite score should be described and whether PCA weights or unit weighting will be used.
- ✓ Wealth indices for both NCSS 2012 and UHS 2013 were computed by data providers, and we have updated the document to make this clear. For NCSS 2012: "The wealth index was generated by data provider using source of drinking water, type of toilet facility, cooking fuel used, lighting type at night, material used to construct floor, wall and roof of dwelling, and household possessions (ownership of household items)". For UHS 2013: "Wealth index was originally constructed by data provider using principal components analysis (PCA) based on the following variables: dwelling characteristics such as presence of electricity, type of water source, type of toilet, and floor, wall, and roof material, household ownership of selected assets and durable goods (radio, television,

motorcycle, computer, refrigerator, electric fan, and automobile), and two indicators of housing tenure (whether the household held title to the dwelling and/or the land)".

- Sample inclusion and exclusion criteria for the proposed project/analyses should be clearly articulated. [EK1]
- ✓ We have included the following sentences: For NCSS 2012 "The sample to be included in the survey was calculated based on the percentage of under-five children with diarrhea in the two weeks preceding to the survey using a margin of error of 0.03, design effect of 1.50 and critical value of $\alpha=0.05$ ". For UHS 2013
- ✓ In addition, we have included the following section in the paper: "Inclusion and exclusion criteria. The analysis will be based on complete case analyses. The choice of variables to be included in constructing intersectional strata for each health condition will be selected by undertaking univariate analyses (54). Only variables that will be significant ($p<0.05$) in the univariate analyses will be used to construct intersectional strata with any correlations assessed using Cramér's V to avoid multicollinearity (55).

- A related point is whether or not any compensation was provided to participants
- ✓ Included in the document: Participation was voluntary, and no compensation or financial incentive was offered for both NCSS 2012 and UHS 2013

- Including conceptual models is a strength. The authors have included a figure/diagram for their analyses. However, the diagram does not capture the complexities of intersectionality outlined in the introduction. There are no indications of what the interactions across levels are, nor are potential mechanisms shown.
- ✓ We have revised Figures 1 and 2 to capture these mechanisms. However, it is important to note that we cannot represent using directed acyclic graphs (DAGs) because we are investigating associations between outcomes and predictors and not their causal relationships.

- Mechanisms are described: eg "their higher prevalence is caused by poor characteristics of physical environment such as poor water drainage, inadequate access to safe water, open sewers, and overcrowding." However, the figure and analyses are not capturing these mechanisms.
- ✓ We have moved that sentence into introduction section and changed "caused" into "association" because we will be looking at associations and not causal effects. The sentence now reads: "The poor hygiene practices in slums which are associated with poor water drainage, inadequate access to safe water, open sewers, and overcrowding also exacerbates health conditions for under-five".

- The MAIHDA approach is clearly sophisticated modelling needed to capture the complex phenomena inherent in intersectionality. The authors have a full description in the appendix appropriate to the statistician. It is a challenge to make the description accessible for the non-statistician. I think the text could be improved by a more step-by-step approach to model building and how to interpret the findings. For example, why is discriminatory accuracy important and what does it tell us in relation to the substantive research questions?
- ✓ We have updated statistical to capture steps involve and now reads. "Using MAIHDA approach, we will capture the unique interaction/intersectional effect for each social group/stratum (i.e., social strata -specific differences in child's diarrhea) while accounting for sample size differences for each social stratum/group by fitting two successive multilevel logistic regression models (44, 45). The first model 1 will be used to assess whether there is significant clustering within intersectional strata (Supporting information Eq. 6). Model 1 will not include any predictor variables and will only have an intercept to estimate the mean health condition and a random effect to model intersectional strata differences (i.e., variance). Model 2 will be used to explore to which extent intersectional strata

differences will be explained by SDoH used in constructing intersectional groups. Model 2 will be an extension of model 1 and will involve adjusting for variables used in constructing intersectional strata as fixed effects (Supporting information Eq. 7). Fixed effects in model 2 will be used to estimate model regression coefficients which will be presented as odds ratio and will describe the association between SDoH variables and under-five health conditions. In the absence of intersectional strata differences, fixed effects used to construct strata are expected to completely explain intersectional strata differences obtained in model 1 implying that strata random effects in model 2 will be equal to zero.

✓ In addition, we have provided a summary of steps in supporting information: Summary of steps to be involved in fitting MAIHDA model

1. Selecting the variables to be used in creating intersectional strata.
2. Creating intersectional strata.
3. Fitting a multilevel model 1 which contains only the random intercept for intersectional strata with no fixed effects.
4. Fitting a multilevel model 2 by including variables used in constructing intersectional strata as fixed effects in model 1.
5. Compute measures of discriminatory accuracy (i.e., variance partitioning coefficient (VPC) and area under the receiver operating characteristic curve (AUC-ROC)) both model 1 and 2.
6. Use VPC for models 1 and 2 to compute proportional change in variance (PCV).

- The description of the MAIHDA approach that is hard to follow is the between vs within level interactions. The description appears to be looking at all multi-way interactions for all predictors. What is the theoretical rationale for testing all interactions vs specific within and between level interactions?

✓ This rationale is described in the following paragraph and expounded in the subsequent paragraphs: The advantage of MAIHDA is that we will look at intersectionality as a mix of both marginalisation and privilege (44, 46, 50). Generally, an interaction-based fixed effects approach looks at intersectionality from the perspective of marginalisation only, which runs the risk of reinforcing the notion of social dominance of the privileged groups which are used as “default” categories. In addition, from an analytical perspective, multilevel models do not face the issues of scalability (i.e., a model’s inability to accommodate an increase in the number of variables included), model parsimony (i.e., a simple model not having great explanatory predictive power), and reduced sample size in some intersectional groups (which influences whether an effect size is determined or not) (45, 51).

- Finally, it would be important to tie the analyses back to issues of inequity outlined in the introduction. A particular challenge may be understanding specific inequities within a population for whom all individuals are marginalized.

✓ This is explained based on the rationale we have described in the above comment. First people living in slums are already marginalised in terms of accessing services such as decent housing and health due to informality of slums. Still in this already marginalised population we have vulnerable groups due to various reasons such as being elderly, disabled, young etc making them more susceptible to worse health. The purpose of these analyses is to unmask these vulnerable groups among under-five based on their SDoH. This is captured in this sentence “This involves treating social strata/groups defined by child’s age and sex, head of household sex, age and education and household’s health insurance status as strata which will used to explain whether health inequalities are shaped by different characteristics in each stratum.

Minor

- “A total of 2,199 children’s data aged under 5 years was provided in the women’s questionnaire.” – shouldbe “data...were”.

✓ Updated “A total of 2,199 children’s data aged under 5 years were provided in the women’s questionnaire.”

ETHICS

- A statement related to ethics approval should be provided.
- ✓ Updated: Ethics and dissemination. The study will use secondary data from the Nairobi Cross-sectional Survey 2012 (NCSS 2012) and Urban Health Survey (UHS 2013) which excludes any participant identifiers. Ethical approval for the NCSS 2012 study was obtained from the Kenya Medical Research Institute's Ethics Review Committee. For UHS 2013, ethical approval was obtained from the Bangladesh Medical Research Council (BMRC) and the Institutional Review Board (IRB) at the School of Public Health, University of North Carolina at Chapel Hill (45). This work as part of ARISE will be used in shaping actions to improve slum health for under five in Bangladesh and Kenya. Finding from these studies will be in published peer reviewed journals and presented in international conferences. Analyses will be presented to policy makers and stakeholders of slum health throughout the course of ARISE project.

REFERNCES

- A number of references appear to be missing elements. I saw the following, but all references should be reviewed.
 - 18. Organization WH. A conceptual framework for action on the social determinants of health.2010.
 - 19. Green MA, Evans CR, Subramanian SV. Can intersectionality theory enrich population health research? 2017.
 - 20. López N, Gadsden VL. Health inequities, social determinants, and intersectionality. 2016.
 - 23. . !!! INVALID CITATION !!! (11, 12).
 - Also see the appendix {Merlo, 2018 #375}
- ✓ Updated

Reviewer: 4

Dr. Janna Metzler, Columbia University

Comments to the Author:

Dear Authors,

I want to thank you for contributing this protocol to the literature and further examining health disparities to support the health and wellbeing of children in urban informal settlements. I've highlighted below a few areas for further clarification that I found challenging when reading through the protocol.

Objective

- Indeed, an intersectional lens on disease burden will lend itself to a more holistic understanding of how to intervene within the population. However, to me, the intent of the analysis is to also provide recommendations for how to intervene to reduce incidence of health conditions noted in the data section. The connection to practice is relevant for readers and can be considered by the authors for its inclusion.
- ✓ This is captured in two ways: Introduction: "Moreover, an intersectional approach enables will us to understand the drivers of health inequities in the context of how individual identities interact with SDoH to promote/rectify health inequalities in dynamic ways in among under-five living in slums". Statistical methods: "This involves treating social strata/groups defined by child's age and sex, head of household sex, age and education and household's health insurance status as strata which will used to explain whether health inequalities are shaped by different characteristics in each stratum".
- Data. It would be helpful to clarify why NCSS 2012 and UHS 2013 household survey data were not used or relevant to the analysis.

✓ We are using these datasets for our proposed analyses as stated in the protocols data section. “These studies will use cross-sectional data from the Nairobi Cross-section Slums Survey (NCSS 2012) for Kenya and Urban Health Survey 2013 (UHS 2013) for Bangladesh, as these are the current disaggregated datasets for slum surveys in both Bangladesh and Kenya”.

- Further explanation as to why other predictors were not included in the analysis may also be relevant for readers – including mother’s age of first marriage (where there is evidence on child marriage as being interconnected to their offspring’s health outcomes); marital status or husband occupancy in the household connected to female-headed household vulnerabilities; etc.

✓ We have included age of mother in the list. In this study context, age of mother is better suited compared to the age of first marriage so that we can capture those children from single headed households. Table 1 shows the list of updated variables where marital status will also be considered for Bangladesh analyses.

VERSION 2 – REVIEW

REVIEWER	Reid, Graham University of Western Ontario, Psychology & Family Medicine
REVIEW RETURNED	01-Apr-2022

GENERAL COMMENTS	Title: Effects of social determinants on children’s health in informal settlements in Bangladesh and Kenya through an intersectionality lens – a study protocol Comments: This is a resubmission of a manuscript I previously reviewed. The authors have made numerous changes and provided a point-by-point response to the reviewers’ comments. The revisions have strengthened the manuscript. However, there are still concerns with this manuscript that should be addressed. Some of the responses to concerns I raised with the first manuscript still remain. In addition, there were multiple elements referred to in the response to reviews that did not appear in the revised manuscript. Key elements referred to (eg Figures) were completely missing. The authors need to do a careful review of the manuscript before resubmission. Additional concerns are summarized below, identified as either “Major” or “Minor” concerns. ABSTRACT I still think a description of the sample and what is being assessed should be added. The added sentence - “The analysis sample will be based on complete case analyses with variables to be included in the analyses will be selected via univariate analyse. – could be cut. Mention key demographics such as sample size, sex distribution. At least state how the primary outcome was assessed. INTRODUCTION Major I previously stated: In the paragraph “An intersectionality lens provides....” the sentence “This is achieved by using multilevel analysis of individual heterogeneity....” seems out of place. I think this should be removed, as this is better placed in the data analyses section. The authors’ response was that they “are also contributing methodologically by applying MAIHDA approach in slum health studies which necessitated the need to mention the
--

	approach in this section". The length and details of the data analyses section and the appendix reflect this goal. However, they need a paragraph in the introduction to support the novelty and importance of this contribution to the literature. The authors have added a paragraph on the rationale for the choice of variables being measured. The phrase "...and discussions with researchers from where the data were collected" - is confusing and is not really a rationale. Either cut this phrase or describe the process and purpose of these discussions. How and why did the researchers suggest the variables? My previous concern still remains: In the Objectives subsection, what "health conditions" refers to should be made explicit, with the rationale for the choices of health conditions measured reviewed in the introduction. First, the text in the response to reviewers doesn't appear in the manuscript. Second, why study diarrhea, fever, cough, and ARI? They have added a sentence "The health conditions which will be considered include diarrhea, cough, fever, and acute respiratory infection (ARI) which mostly affect under-five children in slums". But this doesn't explain why they selected these outcomes. They mention the high prevalence of diarrhea in this same paragraph but say nothing about cough, fever, and acute respiratory infection (ARI). METHODS Major The following sentence is not clear - "The sample was calculated based on the percentage of all births in the three years preceding the date when the survey was delivered using a margin of error of 0.03 which is equivalent to 24% relative difference (95% CI of 20.6-27.4)". Do you mean this was the rationale for the target sample size? I don't follow how a margin of error relates to a relative difference. Also, a "relative difference" between what groups? Minor "A total of 2,199 children's data aged under 5 years was provided in the women's questionnaire." Reword something like: The 4,420 women participants provided data on 2,199 children age 5 and under. "ARI is not available in the (NCSS) 2012 and will therefore not be considered" – could be said more simply, something like: Data on ARI were not available in the NCSS 2012. Something is off here - "Chapel Hill, USA, icddr,b and..." Not sure why parentheses are used - "the (UHS) 2013". Why not just UHS 2013? ANALYSES Major The authors added a rationale for analyzing the datasets separately, but more details are needed. "This is informed by the differences in health priorities and contexts across cities.." – what are the differences in health priorities and contexts? Table 1 is helpful but could be improved. Define NCSS and UHS so that the table can stand alone. Add subcategories for the left
--	--

	hand column to differentiate outcomes vs predictors. Consider adding additional details such as the questions asked for each variable and/or coding. One option would be to have a supplementary table listing the exact wording of items and response options, with Table 1 being the variable and coding. This is particularly important for variables that are uncommon (e.g., income generating activity) or have labels that are someone ambiguous (e.g., tenure). My previous point “Each of the variables should be described in detail.” has not been addressed. The above suggested revisions to Table 1 would provide a fuller description. I suggested specifying the inclusion and exclusion criteria. The text in the letter - “For NCSS 2012 “The sample to be included in the survey was calculated based on the percentage of under-five children with diarrhea in the two weeks preceding to the survey using a margin of error of 0.03, design effect of 1.50 and critical value of $\alpha=0.05$”. For UHS 2013” – is not in the manuscript and this response is incomplete as there is nothing after “For UHS 2013”. That said, it is unclear to me why the inclusion criterion is the “percentage of under-five children with diarrhea in the two weeks preceding the survey”. The aim of the study was not to estimate the prevalence of diarrhea. Finally, the other text added in the subsection “Inclusion and exclusion criteria”, does not actually discuss the inclusion and exclusion criteria. Revised Figures 1 and 2. I disagree with the authors’ view that because they are testing associations, they cannot present theoretical causal pathways in a figure. I continue to find the description of MAIHDA confusing and the links to the theory/conceptual models hard to follow. “MAIHDA aims to primarily identify intersectional effects by identifying groups that are advantaged than would be expected in the absence of interaction by distinguishing between additive and interaction effects (28, 51).” – I think some words are missing from this sentence. The sentences from “MAIHDA aims to primarily identify intersectional effects” to “....additional intersectional effect specific to that group.” contains multiple elements. Consider breaking into multiple sentences and explaining the link between the theory and the analysis. In the letter it states a “Summary of steps to be involved in fitting MAIHDA model “ is included. This sounds good but is not in the manuscript. If they authors are referring to this list of steps in the appendix, this is inadequate. Consider walking the reader through each of these steps in the manuscript. Minor “Wealth index was constructed by data provider using” – unclear what “by data provider” means. Check that the term “informal settlements” has been replaced throughout the entire manuscript.
--	---

	“sex and each of the other five variables and not all interactions(54) Eq. 3.” – what is Eq. 3?
--	---

VERSION 2 – AUTHOR RESPONSE

Response to reviewer

Title: Effects of social determinants on children’s health in informal settlements in Bangladesh and Kenya through an intersectionality lens – a study protocol

Comments: This is a resubmission of a manuscript I previously reviewed. The authors have made numerous changes and provided a point-by-point response to the reviewers’ comments. The revisions have strengthened the manuscript. However, there are still concerns with this manuscript that should be addressed. Some of the responses to concerns I raised with the first manuscript still remain. In addition, there were multiple elements referred to in the response to reviews that did not appear in the revised manuscript. Key elements referred to (eg Figures) were completely missing. The authors need to do a careful review of the manuscript before resubmission. Additional concerns are summarized below, identified as either “Major” or “Minor” concerns.

✓ Thank you for reviewing the manuscript again and we greatly appreciate. We have addressed all comments and revisited some concerns raised in the first manuscript.

ABSTRACT

I still think a description of the sample and what is being assessed should be added. The added sentence - “The analysis sample will be based on complete case analyses with variables to be included in the analyses will be selected via univariate analyse. – could be cut. Mention key demographics such as sample size, sex distribution. At least state how the primary outcome was assessed.

We have cut the univariate analyses sentence. In addition, we have revised methods and analysis section to read as follows:

The protocol describes how we will analyse data from the Nairobi Cross-sectional Slum Survey (NCSS 2012) for Kenya and the Urban Health Survey (UHS 2013) for Bangladesh to explore how SDoH influence under-five health conditions in slums within an intersectionality framework. The NCSS 2012 and UHS samples will consist of 2,199 and 3,173 under-fives, respectively. The analysis sample will be based on complete case analyses with variables to be included in the analyses will be selected via univariate analyses. We will apply Multilevel Analysis of Individual Heterogeneity and Discriminatory Accuracy (MAIHDA) approach. Some of SDoH characteristics to be considered will include those of children, head of household, mothers, and social structure characteristics of household. The primary outcome outcomes will be whether a child had measures will be diarrhea, cough, fever, and acute respiratory infection (ARI) two weeks preceding surveys.

At least state how the primary outcome was assessed.

The primary outcome outcomes will be whether a child had measures will be diarrhea, cough, fever, and acute respiratory infection (ARI) two weeks preceding surveys.

INTRODUCTION

Major

I previously stated: In the paragraph “An intersectionality lens provides....” the sentence “This is achieved by using multilevel analysis of individual heterogeneity....” seems out of place. I think this should be removed, as this is better placed in the data analyses section. The authors’ response was that they “are also contributing methodologically by applying MAIHDA approach in slum health studies which necessitated the need to mention the approach in this section”. The length and details of the data analyses section and the appendix reflect this goal. However, they need a paragraph in the introduction to support the novelty and importance of this contribution to the literature.

This is achieved by using multilevel analysis of individual heterogeneity....” seems out of place.

We have removed the sentence and incorporated its information into methods section and replaced it with:

Quantitatively this can be explored using Multilevel Analysis of Individual Heterogeneity and Discriminatory Accuracy (MAIHDA) approach.

However, they need a paragraph in the introduction to support the novelty and importance of this contribution to the literature.

This is answered by this sentence

This approach will enable us to understand the drivers of health inequities in the context of how individual identities interact with SDoH to promote/rectify health inequalities in dynamic ways in among under-five living in slums quantitatively

The authors have added a paragraph on the rationale for the choice of variables being measured. The phrase “...and discussions with researchers from where the data were collected” - is confusing and is not really a rationale. Either cut this phrase or describe the process and purpose of these discussions. How and why did the researchers suggest the variables?

The phrase “...and discussions with researchers from where the data were collected” We have cut it out.

My previous concern still remains: In the Objectives subsection, what “health conditions” refers to should be made explicit, with the rationale for the choices of health conditions measured reviewed in the introduction. First, the text in the response to reviewers doesn’t appear in the manuscript. Second, why study diarrhea, fever, cough, and ARI? They have added a sentence “The health conditions which will be considered include diarrhea, cough, fever, and acute respiratory infection (ARI) which mostly affect under-five children in slums”. But this doesn’t explain why they selected these outcomes. They mention the high prevalence of diarrhea in this same paragraph but say nothing about cough, fever, and acute respiratory infection (ARI).

To make it clear we have replaced “health conditions” with “health outcomes” throughout the manuscript.

.First, the text in the response to reviewers does not appear in the manuscript.

We reviewed as follows” The primary analytic objective is to systematically examine how various SDoH, and individual factors affect health conditions (i.e., diarrhea, fever, cough, and ARI) of under-five children in slums within an intersectionality framework. Now it reads “The primary analytic objective is to systematically examine how various SDoH, and individual factors affect health outcomes (i.e., diarrhea, fever, cough, and ARI) of under-five children in slums within an intersectionality framework.”

“The health conditions which will be considered include diarrhea, cough, fever, and acute respiratory infection (ARI) which mostly affect under-five children in slums”. But this doesn’t explain why they selected these outcomes. They mention the high prevalence of diarrhea in this same paragraph but say nothing about cough, fever, and acute respiratory infection (ARI).

We have added the following sentence in the introduction section: For example, poor sanitation and lack of safe drinking water makes diarrhea a leading cause of death among children aged below five years, while fevers, coughs, and acute respiratory infections (ARI) are mainly caused by the poor state of housing and overcrowding.

METHODS

Major

The following sentence is not clear - “The sample was calculated based on the percentage of all births in the three years preceding the date when the survey was delivered using a margin of error of 0.03 which is equivalent to 24% relative difference (95% CI of 20.6-27.4)”. Do you mean this was the rationale for the target sample size? I don’t follow how a margin of error relates to a relative difference. Also, a “relative difference” between what groups?

We have revised to: The key indicators used to calculate the sample size were (i) under-five mortality and (ii) percentage of birth deliveries in the health facilities for all births in the last three years.

Minor

“A total of 2,199 children’s data aged under 5 years was provided in the women’s questionnaire.”
Reword something like: The 4,420 women participants provided data on 2,199 children age 5 and under.

Done: The 4,420 women participants provided data on 2,199 children aged 5 and under.

“ARI is not available in the (NCSS) 2012 and will therefore not be considered” – could be said more simply, something like: Data on ARI were not available in the NCSS 2012. Something is off here -

“Chapel Hill, USA, icddr,b and...”

Revised into: Data on ARI and diarrhea were not available in the NCSS 2012 and UHS 2013, respectively.

Not sure why parentheses are used - “the (UHS) 2013”. Why not just UHS 2013?

Revised into UHS 2013

ANALYSES Major

The authors added a rationale for analyzing the datasets separately, but more details are needed. “This is informed by the differences in health priorities and contexts across cities..” – what are the differences in health priorities and contexts?

We have added “The underlying social and living conditions in Dhaka and Nairobi slums are different which necessitates two distinct publications for Bangladesh and Kenya for effective interventions” in data section. In addition, there is this sentence “The differences in outcome and explanatory variables in NCSS 2012 and UHS 2013 also informed the need for separate analyses for Kenya and Bangladesh”

Table 1 is helpful but could be improved. Define NCSS and UHS so that the table can stand alone. Add subcategories for the left-hand column to differentiate outcomes vs predictors. Consider adding additional details such as the questions asked for each variable and/or coding. One option would be to have a supplementary table listing the exact wording of items and response options, with Table 1 being the variable and coding. This is particularly important for variables that are uncommon (e.g., income generating activity) or have labels that are someone ambiguous (e.g., tenure).

We have revised Table 1 and in addition we have included Table s1 for NCSS 2012 and Table s2 for UHS 2013 in supplemental material which contains variable name, description, and categories.

My previous point “Each of the variables should be described in detail.” has not been addressed. The above suggested revisions to Table 1 would provide a fuller description.

This has been addressed by including Tables s1 and s2 in supplemental material.

I suggested specifying the inclusion and exclusion criteria. The text in the letter - “For NCSS 2012 “The sample to be included in the survey was calculated based on the percentage of under-five children with diarrhea in the two weeks preceding to the survey using a margin of error of 0.03, design effect of 1.50 and critical value of $\alpha=0.05$ ”. For UHS 2013” – is not in the manuscript and this response is incomplete as there is nothing after “For UHS 2013”. That said, it is unclear to me why the inclusion criterion is the “percentage of under-five children with diarrhea in the two weeks preceding the survey”. The aim of the study was not to estimate the prevalence of diarrhea.

We have revised this to read “The sample to be included in the survey was calculated based on the percentage of children 12-23 months who had been fully immunized using a margin of error of 0.03, design effect of 1.50 and critical value of $\alpha=0.05$ (8). The number of households required to estimate the percentage of children 12-23 months fully vaccinated was large enough to allow estimation of the other indicators such as diarrhea, fever and cough with the specified precision.

Finally, the other text added in the subsection ““Inclusion and exclusion criteria”, does not actually discuss the inclusion and exclusion criteria.

We have removed this subsection from manuscript since inclusion and exclusion is covered in data section.

Revised Figures 1 and 2. I disagree with the authors’ view that because they are testing associations, they cannot present theoretical causal pathways in a figure.

We have revised and included only Figure 1 indicating causal pathway between the four categories of variables and health outcomes.

Figure 1: A causal diagram showing the direct pathways between children’s demographics, child’s mother characteristics, head of household demographics, social structure characteristics and under-five health conditions.

I continue to find the description of MAIHDA confusing and the links to the theory/conceptual models hard to follow.

We have revised this extensively where we have included the two main steps of MAIHDA approach. Step1 which includes fitting of null model in which individuals at level one are nested within social groups at level two; while step 2 which involves extending null model by adjusting for variables used in constructing intersectional strata as fixed effects.

We have moved illustrative example to supplemental material from the main document.

“MAIHDA aims to primarily identify intersectional effects by identifying groups that are advantaged than would be expected in the absence of interaction by distinguishing between additive and interaction effects (28, 51).” – I think some words are missing from this sentence.

We have revised this into “MAIHDA aims to primarily identify intersecting inequalities in a quantitative way by defining intersectional groups according to combinations of social attributes which is like clustering of individuals based on some shared attributes such as neighborhood, school, or household, among others.

The sentences from “MAIHDA aims to primarily identify intersectional effects” to “....additional intersectional effect specific to that group.” contains multiple elements. Consider breaking into multiple sentences and explaining the link between the theory and the analysis.

Revised into” “MAIHDA aims to primarily identify intersecting inequalities in a quantitative way by defining intersectional groups according to combinations of social attributes.

In the letter it states a “Summary of steps to be involved in fitting MAIHDA model “ is included. This sounds good but is not in the manuscript. If they authors are referring to this list of steps in the appendix, this is inadequate. Consider walking the reader through each of these steps in the manuscript.

We have summarised steps into main document into two steps where the first step is fitting a null model with no fixed effects but only a random intercept of intersectional strata. The second step involves fitting model 2 which is an extension of null model by including variables used in constructing intersectional strata as fixed effects.

In addition, we have moved the illustrative example into supplemental material and related the 2 steps in the main document with it.

Minor

“Wealth index was constructed by data provider using” – unclear what “by data provider” means.

Revised to read “The wealth index was generated using source of drinking water, type of toilet facility, cooking fuel used, lighting type at night, material used to construct floor, wall and roof of dwelling, and household possessions (ownership of household items)

Check that the term “informal settlements” has been replaced throughout the entire manuscript.

Done

“sex and each of the other five variables and not all interactions(54) Eq. 3.” – what is Eq. 3?

Revised and that paragraph into supporting document. It implied Equation 3

VERSION 3 – REVIEW

REVIEWER	Reid, Graham University of Western Ontario, Psychology & Family Medicine
REVIEW RETURNED	12-May-2022

GENERAL COMMENTS	This is the third submission of a manuscript I previously reviewed. The authors have again made numerous changes and provided a point-by-point response to the reviewers’ comments. Overall, I feel the response to my comments and revisions have address my suggestions/concerns. The manuscript describes both the rationale for the topic being studied and the novel data analytic approach being used.
--